# Growing up with Idiopathic Pulmonary Arterial Hypertension: An Arduous Journey

Tanguy Dutilleux [1], Nesrine Farhat [2], Ruth Heying [3], Marie-Christine Seghaye [2,*] and Maurice Beghetti [4]

1 Department of Pediatric Cardiology, University Hospital Aachen, 52074 Aachen, Germany
2 Department of Pediatric Cardiology, Centre Hospitalier Universitaire de Liège, 4000 Liège, Belgium
3 Department of Pediatric Cardiology, University Hospital Leuven, 3000 Leuven, Belgium
4 Department of Pediatric Cardiology, Children's University Hospital Geneva, 1205 Geneva, Switzerland
* Correspondence: mcseghaye@chuliege.be; Tel.: +32-4-323-9275

**Abstract:** Idiopathic pulmonary arterial hypertension (IPAH) is an uncommon and severe disease. We report the case of a 7-year-old boy investigated for cardiac murmur and exercise intolerance. Pulmonary hypertension (PH) was suspected at clinical examination and confirmed by echocardiography and cardiac catheterization. This case of pulmonary hypertension was classified as idiopathic given the negative etiological investigation. Vasoreactive testing with oxygen and nitric oxide was negative. Therefore, treatment with sildenafil (1.4 mg/kg/d) and bosentan (3 mg/kg/d) was initiated. This allowed the stabilization of, but not a decrease in, pulmonary artery pressure for the next 5 years, during which the patient's quality of life was significantly reduced. At a later follow-up, the estimated pulmonary pressure was found to have increased and become supra-systemic, with a consequent deterioration in the child's condition. This led to the decision to enter him into a clinical trial that is still ongoing. Idiopathic pulmonary arterial hypertension is a severe disease that can present with non-specific symptoms, such as asthenia and exercise limitation, which are important not to trivialize. The disease is associated with significantly decreased quality of life in affected children and carries a high burden in terms of mortality and morbidity. The current knowledge about IPAH in children is reviewed, with a particular focus on the future prospects for its treatment and the related quality of life of patients.

**Keywords:** pulmonary arterial hypertension; idiopathic; children; vasoreactivity; treatment; quality of life

## 1. Introduction

Pulmonary hypertension is a rare but severe condition defined by a mean pulmonary arterial pressure (mPAP) > 20 mmHg at rest [1].

Depending on the etiology, five different groups of PH are described and listed in Table 1 [1,2]: pulmonary arterial hypertension (PAH) (group 1), PH associated with left heart disease (group 2), PH associated with lung disease and/or hypoxemia (group 3), PH associated with pulmonary artery obstruction (group 4), and PH with unclear and/or multifactorial mechanisms (group 5) [1–4].

PAH or pre-capillary pulmonary hypertension, which includes idiopathic pulmonary hypertension (IPAH), is defined as pulmonary hypertension with a pulmonary-artery-wedge pressure ≤ 15 mmHg with PVR ≥ 2 WU (2), excluding groups 4 and 5.

Idiopathic pulmonary arterial hypertension is a rare disease with an incidence ranging between 0.47 and 1–2 cases per million children, and with a prevalence of 2.1 to 4.4 cases per million children [2].

Although most children with PAH can now reach adulthood, thanks to recent therapeutic advances, the burden created by the disease in terms of morbidity and mortality remains high [5].

Pulmonary hypertension in children is associated with decreased health-related quality of life due, among other factors, to exercise intolerance, social isolation, and the side effects of medications [6].

The diagnosis of PAH is suggested upon rigorous clinical examination and requires confirmation by echocardiography and, ultimately, by cardiac catheterization with hemodynamic investigation, which allows the precise measurement of PVR and to test vasoreactivity through the inhalation of pulmonary vasodilators, such as oxygen or nitric oxide. The results of this test determine the treatment choice and prognosis [1,3].

We report a rare case of IPAH in a young boy and discuss the current recommendations in terms of exploration and treatment.

The delayed incidental diagnosis of this case should make pediatricians aware of the necessity of investigating exercise intolerance.

## 2. Case Report

A 7-year-old boy was referred to the cardiology-outpatient clinic because of a cardiac murmur.

His familial history was uneventful, as had been his own until the beginning of his schooling 2 years before his presentation. From this point onward, the only clinical sign observed by his parents was his limited exercise tolerance compared with that of his peers.

At clinical examination, the boy was eutrophic. His blood pressure and transcutaneous oxygen saturation ($SaO_2$) were normal. In addition to a hyperactive precordium, an intense second sound was heard, along with a diastolic murmur over the pulmonary artery. There was no sign of cardiac insufficiency. The pulmonary auscultation was normal. The liver was not enlarged.

Electrocardiography (ECG) showed right ventricular (RV) hypertrophy.

Echocardiography excluded a structural cardiac defect and confirmed pulmonary hypertension with dilated right atrium (RA), dilated and hypertrophied RV, and dilated main and peripheral pulmonary arteries. A tricuspid regurgitation grade of 3/4 was also shown, which allowed the estimation of the systolic RV pressure at a systemic level of 80 mmHg. A pulmonary regurgitation grade II was noticed, which allowed the estimation of the diastolic pulmonary pressure at 48 mmHg.

Examinations were carried out to exclude autoimmune process, such as systemic lupus erythematosus, mixed connective-tissue disease and systemic sclerosis, pro-thrombotic disease, obstructive apnoea syndrome, pulmonary emboli, and interstitial pulmonary diseases, such as childhood interstitial lung disease.

However, microcytic anemia due to minor β-thalassemia was detected.

The NT-proBNP level was slightly elevated (142 ng/L).

Upon genetic testing, bone morphogenic protein receptor (BMPR)-2 mutation was excluded. Further gene mutations have not yet been investigated.

The patient's 6-min walking distance was 320 m (Z-score: $-3.5$), with the $SaO_2$ decreasing to 90% at the end of the test.

Cardiac catheterization allowed the measurement of the cardiac output according to the Fick principle and confirmed elevated pulmonary vascular resistance (PVR), which slightly decreased upon the inhalation of 100% oxygen ($O_2$) and nitric oxide (NO) but did not reach the vasoreactivity threshold. The mean pulmonary artery pressure (PAP) was 60 mmHg (systemic level), with a normal pulmonary-capillary-wedge pressure (9 mmHg). The PVR, calculated while the patient breathed ambient air, was 10 WU, and dropped to 6.5 WU during vasoreactive testing with oxygen and nitric oxide.

Table 1 summarizes hemodynamic parameters before and during the vasoreactive testing.

Treatment using the phosphodiesterase 5 inhibitor (PDE-5) sildenafil ($3 \times 10$ mg/d) and the endothelin receptor antagonist (ERA) bosentan ($2 \times 31.25$ mg/d) was introduced. The clinical evolution was stable for the first 5 years, while the bosentan dosage was

adapted to the patient's weight (2 × 62.5 mg/d). However, the systolic PAP, estimated by echocardiography, remained elevated.

**Table 1.** Hemodynamic parameters before and during vasoreactive testing with oxygen ($O_2$) and nitric oxide (NO).

|  | Baseline ($O_2$ 21%) | $O_2$ 100% + NO 16 ppm |
|---|---|---|
| PAP (s/d-m) mmHg | 75/47–60 | 64/38–51 |
| AoP (s/d-m) mmHg | 77/49–60 | 59/41–48 |
| PCWP mmHg | 9 | Not measured |
| CO L/min (Fick Formula) | 1.8 | 2.1 |
| CI L/min/m$^2$ | 5.0 | 7.8 |
| PVRI WU | 10 | 6.5 |
| PAPm/AoPm | 1 | 1.1 |

Note: PAP: pulmonary arterial pressure; AoP: aortic pressure; PCWP: pulmonary capillary wedge pressure; CO: cardiac output; CI: cardiac index; PVRI: pulmonary vascular resistance; s: systolic; d: diastolic; m: mean. ppm: part per million.

Clinically, the patient was in a relatively stable condition, but was no longer able to participate in social activities or school sport sessions, which led him to experience social isolation. This was accentuated by the protective attitude of his family.

At the age of 12 years, the patient's RV systolic pressure increased up to supra-systemic values of 130 mmHg. This went along with a significant increase in NT-Pro-BNP of 998 ng/L and the occurrence of chest pain. The bosentan dose was doubled, which led to clinical improvement, but without an effect on RV pressure or function. At this point, there was no argument in favor of the loss of therapeutic compliance.

The introduction of prostacyclin was not considered. Instead, the patient was selected for inclusion in an experimental trial. The performance of aorto-pulmonary Pott's anastomosis was postponed.

Table 2 summarizes the clinical course and echocardiographic parameters of the patient over a timeline of 5 years.

**Table 2.** Clinical course and echocardiographic parameters over a timeline of 5 years.

|  | Diagnosis | 12 Months | 24 Months | 36 Months | 48 Months | 60 Months |
|---|---|---|---|---|---|---|
| **PVD (mmHg)** | 61 + RAP | 68 + RAP | 76 + RAP | 90 + RAP | 109 + RAP | 130 + RAP |
| **PR (grade)** | 2/6 | 2/6 | 2/6 | 2/6 | 2/6 | 2/6 |
| **dPAP (mmHg)** | 69 | 70 | 70 | 80 | 80 | 80 |
| **EDRVD (mm)** | 25 | 30 | 30 | 35 | 33 | 37 |
| **TAPSE (mm)** |  |  | 16 | 19 | 30 | 20 |
| **LV EF (%)** | 66 | 61 | 65 | 50 | 64 | 81 |
| **6 min WT (m)** | 320 | 309 | 410 | 460 | 417 |  |
| **SaO$_2$ (%)** | 90 | 94 | 92 | 91 | 97 |  |
| **Z-score** | −3.5 | −4 | −2.5 | −2.5 | −3.5 |  |
| **Treatment** |  |  |  |  |  |  |
| **Bosentan (mg/kg/d)** | 2 × 31.25 (3) | 2 × 31.25 (2.4) | 2 × 31.25 (2) | 2 × 31.25 (1.8) | 2 × 62.5 (2.8) | 2 × 62.5 (2.65) |
| **Sildenafil (mg/kg/d)** | 3 × 10 (1.4) | 3 × 10 (1.15) | 3 × 20 (1.9) | 3 × 20 (1.7) | 3 × 20 (1.3) | 3 × 20 (1.27) |
| **Weight (kg)** | 21 | 26 | 31 | 35 | 45 | 47 |
| **NT-proBNP (ng/L)** | 142 |  |  | 189 | 570 | 988 |

Note: RAP: right atrial pressure; RVP: estimated right ventricular pressure; PR: pulmonary regurgitation; dPAP: estimated diastolic pulmonary arterial pressure; EDRVD: end-diastolic right ventricular diameter; TAPSE: tricuspid annular plane systolic excursion; LVEF: left ventricular ejection fraction; WT: walking test. SaO$_2$: transcutaneous oxygen saturation at the end of the test; Z score based on height.

## 3. Discussion

We present the case of a young boy with IPAH and his follow-up period of 5 years.

The initial reason for consultation was the presence of a cardiac murmur. The patient's history revealed chronic exercise intolerance, which is a common complaint in school-aged children and infrequently related to a cardiac disease. The relationship with minor thalassemia appeared to be unlikely. Our case illustrates the importance of exploring exercise intolerance in the pediatric population and of the exclusion of PH.

The clinical examination, during which a loud second sound was heard, was suggestive of PH and should have guided the diagnosis earlier.

This report also points out the difficulties for children living with PAH.

In the recent years, a few studies have focused their work around children with PAH. By assessing their quality of life, using validated tools to compare them with healthy children, it was found that these children have a worse perceived quality of life. Patients with PH had significantly lower scores in all areas of functioning assessed (physical, emotional, social, and academic) [7,8].

This low quality of life due, in particular, to the fact that PAH is related to the limited adaptation of cardiac output. This exerted negative effects on academic and social life in our case, in which the child was excluded from most of the activities in which his peers took part. The potential risk of sudden death and syncope induced major stress among the family.

Furthermore, the treatment itself involves poly-pharmacy, repeated daily oral doses, and even continuous parenteral therapy (prostacyclin therapy), and the side effects of these dugs significantly affect quality of life [7,8]. Despite its heavy treatment, the course of PAH is usually characterized by progressive clinical deterioration, or even death.

Finally, studies have mentioned negative effects of the disease on parents, including increased stress, anxiety, and depression as well [6]. In addition, parents may also develop a protective attitude, which increases social isolation, as it was the case in our patient.

Thus, it seems clear that physical, emotional, social, and academic functioning must be addressed during routine medical care and therapeutic decision making [7].

### 3.1. Idiopathic Pulmonary Hypertension

3.1.1. Clinical Presentation

The clinical signs of PH and all forms of PAH are non-specific, which explains in part why late diagnosis is the rule [3].

As in our patient's case, the most frequent symptoms are fatigue, exercise intolerance, and dyspnea due to the inability to increase cardiac output on demand and to progressive decrease in RV function. Failure to thrive may occur as a consequence of increased caloric expenditure and intestinal malabsorption secondary to right-heart failure [4].

Cyanosis at rest or at exercise is observed in patients with intra-cardiac shunt, most frequently in the form of atrial septum defects with right-to-left shunt at rest or at exercise [2,3]. Chronic cyanosis results in polycythemia, which is in turn responsible for headache and increased risk of thrombosis.

Syncope often occurs post-exercise or in situations related to Valsalva effort, and is always an alarming sign. It is frequently the first manifestation of IPAH in children [2,9].

Finally, there is a risk of supraventricular arrhythmia in children with severe right atrial enlargement. Patients who suffer from supraventricular arrhythmia can present with palpitations and have a higher risk of sudden cardiac death [1,2,10].

At clinical examination, the right ventricle may appear hyperdynamic. Cardiac auscultation may reveal a loud second sound, a holosystolic murmur due to tricuspid regurgitation, and a protodiastolic murmur due to pulmonary regurgitation.

Clinical signs, such as right heart failure with jugular distension, hepatomegaly, ascites, and peripheral edema may also be present [2,3].

### 3.1.2. Diagnosis and Evaluation

Idiopathic pulmonary arterial hypertension is a form of PAH for which no underlying cause has been identified. It remains a diagnosis of exclusion. The diagnosis of PAH requires a careful and extensive workup because PAH is associated with a wide range of potentially treatable diseases.

The personal and family history are of paramount importance, especially a positive family history of PAH, congenital heart disease, and any other congenital malformations, genetic diseases or mutations, rheumatologic disorders, or sudden and unexplained death [2].

The clinical examination should focus on the symptoms and signs described above.

When PAH is suspected, the initial evaluation should at least include an ECG, a chest radiograph, and an echocardiography.

On ECG, there are signs of right ventricular pressure overload [2]. Typically, chest X-rays show non-specific signs, such as cardiomegaly or dilated pulmonary arteries [1,3]. Echocardiography plays an important role in the diagnosis of PAH. Firstly, it allows the exclusion of structural cardiac defect. Second, it is a tool to non-invasively estimate systolic RV pressure (P), which is a surrogate for systolic PAP, in cases in which right-ventricular-outflow-tract obstruction is excluded.

To this end, the peak flow velocity of the tricuspid regurgitation (TR) is measured and the modified Bernoulli formula is applied, where $RVP = 4 \times TR^2$ + estimated right atrial pressure [11,12]. Pulmonary arterial diastolic pressure (PADP) can also be estimated by measuring the pulmonary regurgitation flow velocity (PR) and applying the modified Bernoulli formula, where $PADP = 4 \times$ end diastolic $PR^2$ + estimated right atrial pressure [11,12].

In addition, ultrasound can be used to examine the functions of both ventricles. In both PAH and IPAH, the examination of the right ventricle is fundamental. For instance, the size of the cavity, the thickness of the free wall and the systolic excursion of the tricuspid annular plane (TAPSE) are studied [2]. Furthermore, the determination of the right ventricular end-systolic remodeling index has been shown to be a helpful indicator of RV-pressure overload in children, especially when the RVP cannot be estimated due to technical difficulties in analyzing the TR [12].

As soon as PH is suspected on echocardiography, an etiological investigation should be performed (Table 3). Finally, a hemodynamic study using cardiac catheterization must take place.

Indeed, only cardiac catheterization allows a certain diagnosis. It enables the precise measurement of pulmonary arterial pressure and the calculation of pulmonary arterial resistances [3,9].

This is of central importance for the etiological diagnosis, for the evaluation of the degree of severity, the assessment of potential vasoreactivity, and for the prognosis of the disease.

Cardiac catheterization should include an acute vasodilation test (AVT). This test consists of the administration of a short-acting vasodilator (e.g., inhaled NO). The response to AVT is defined by the Sitbon criteria as a decrease in mPAP $\geq$ 10 mmHg, reaching an mPAP value < 40 mmHg, while cardiac output increases or remains unchanged. Between 10% and 15% of children with IPAH are responders [13,14]. Patients who respond to AVT have a vasoreactive, and usually less severe form of the disease, and should be treated with calcium channel blockers. Unfortunately, our patient did not respond to AVT.

Patients with PAH should also receive genetic counselling to assess the need for a genetic workup, especially if no etiology is identified. Indeed, several gene mutations involved in the pathogenesis of PAH have been identified in 20–30% of sporadic pediatric PAH cases and in 70–80% of familial PAH cases. Two genetic testing options are available: gene-panel testing and whole-genome sequencing. Generally, it is recommended to start with gene-panel testing, which is faster and less expensive, before moving on to genome sequencing if a genetic origin of the disease is strongly suspected [15,16].

**Table 3.** Adapted from [1–3].

| Clinical Classification of Pulmonary Hypertension (PHT) (6th World Symposium) |
| --- |
| (1) Pulmonary arterial hypertension (PAH) |
|     1.1 Idiopathic PAH |
|     1.2 Heritable PAH |
|     1.3 Drug and toxin induced PAH |
|     1.4 Associated with: |
|         A: Connective tissue disease |
|         B: HIV Infection |
|         C: Portal hypertension |
|         D: Congenital heart disease |
|         E: Schistosomiasis |
|     1.5 Long-term PAH responders to calcium channel blockers |
|     1.6 PAH with features of venous/capillaries involvement |
|     1.7 Persistent PHT-of-the-newborn syndrome |
| (2) Pulmonary hypertension due to left-heart disease |
|     2.1 PHT due to heart failure with preserved left ventricular ejection fraction (LVEF) |
|     2.2 PHT due to heart failure with reduced left ventricular ejection fraction |
|     2.3 Valvular heart disease |
|     2.4 Congenital/acquired cardiovascular conditions leading to post-capillary PHT |
| (3) Pulmonary hypertension due to lung disease and/or hypoxia |
|     3.1 Obstructive lung disease |
|     3.2 Restrictive lung disease |
|     3.3 Other lung disease with mixed restrictive/obstructive pattern |
|     3.4 Hypoxia without lung disease |
|     3.5 Developmental lung disorders |
| (4) Pulmonary hypertension due to pulmonary artery obstruction |
|     4.1 Chronic thromboembolic PHT |
|     4.2 Other pulmonary artery obstructions |
| (5) Pulmonary hypertension with unclear and/or multifactorial mechanisms |
|     5.1 Hematologic disorders |
|     5.2 Systemic and metabolic disorders |
|     5.3 Others |
|     5.4 Complex congenital heart disease |

A final important point is the determination of the blood levels of BNP and its precursor NT pro-BNP, which is synthesized by cardiomyocytes in response to stretch. The NT-proBNP is an important biomarker, which suggests a cardiac etiology and allows the evaluation of the severity of the disease, the response to treatment, the progression of the disease and, finally, the risk of mortality [17].

Indeed, patients with BNP values < 50 ng/L have 2-, 3- and 5-year survival rates of 100%, 99%, and 97% respectively. In contrast, those with BNP values > 537 ng/L have significantly poorer survival rates [18].

### 3.1.3. Treatment

Over the past two decades, the prognosis of children with PAH and IPAH has improved due to significant advances in the understanding of the pathophysiology of the disease, and to results emerging from child-based studies, allowing the optimization of therapies [8,11,17].

The treatment of IPAH depends on the ability of the pulmonary arteries to react and dilate when challenged with vasodilatory mediators, such as inhaled NO. This reactivity must be tested invasively through catheterization, while PVR is calculated during the inhalation of NO, as discussed above. If the patient shows a decrease in PVR and meets the Sitbon criteria, he/she is considered as presenting with the vasoreactive form, which usually responds well to calcium channel blockade (CCB). The AVT must be controlled,

given that 45% of patients with an initially positive response become resistant during follow-up (the average follow-up time reported is 10 years) [2].

If there is no decrease in PVR at AVT, the patient is considered to have no vasoreactivity and is considered an AVT-non-responder. For those who do not respond to AVT, the risk stratification must be established to adapt the therapeutic strategy. The level of risk depends on the clinical signs, echocardiographic and hemodynamic measurements, and laboratory examinations, respectively.

Figure 1 summarizes the treatment algorithm, depending on AVT, following the latest recommendations, and Figure 2 presents the risk stratification used to determine optimal therapy [2].

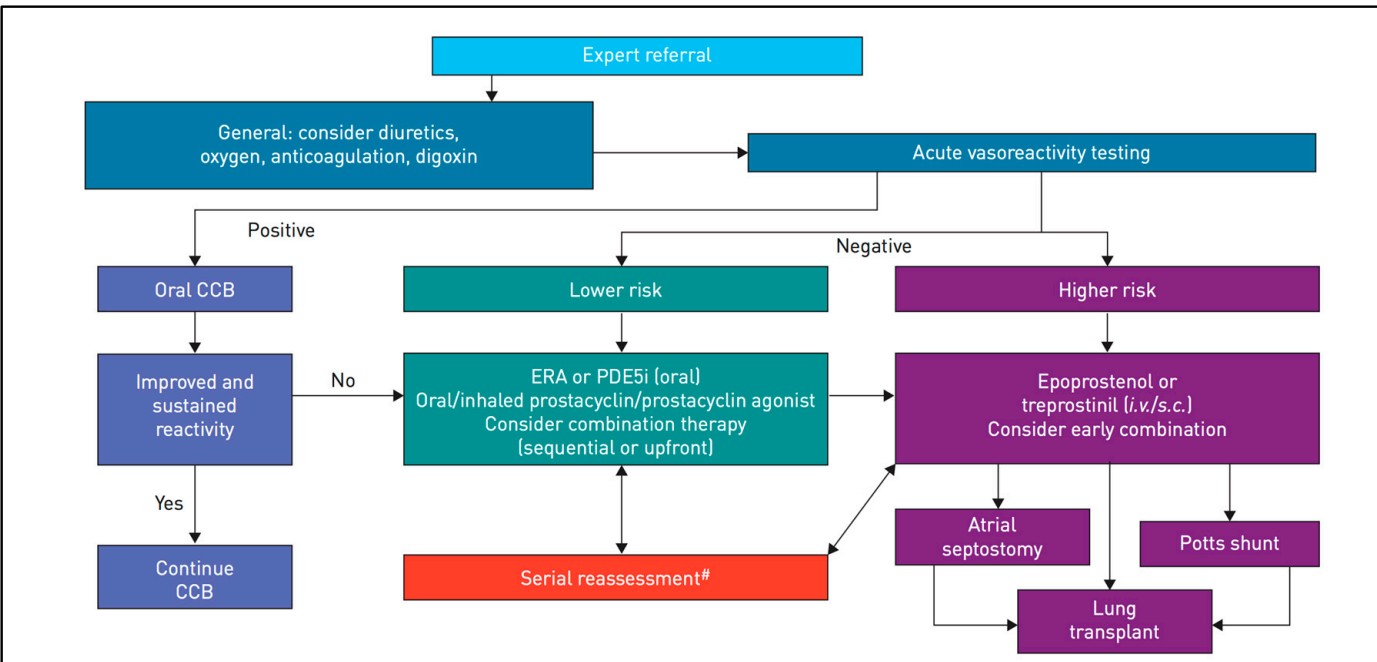

**Figure 1.** Treatment algorithm for familial and/or idiopathic pulmonary arterial hypertension. Adapted from [2]. CCB: calcium channel blocker; ERA: endothelin receptor agonist; PDE5i: phosphodiesterase type 5 inhibitor; #: deterioration or not meeting treatment goals.

For AVT-non-responders at lower risk, it is advisable to start directly with combined therapy. Since our patient was a non-responder with clinical impairment, we started dual therapy with PDE-5 and ERA after cardiac catheterization. In AVT-non-responders at higher risk, the treatment is based on trepostinil or epoprostenol given intravenously or subcutaneously. In cases involving clinical deterioration, lung transplantation and interventional palliative bridges, such as atrioseptostomy or aorto-pulmonary shunts (Pott's shunt) must be considered. The last therapeutic option is heart–lung transplantation [1–3,19]. The latter is indicated for patients with end-stage PAH and signs of irreversible dysfunction of the right ventricle with signs of fibrosis or infarction that are thought to be irreversible [19]. The right ventricle, when subjected to increased pulmonary vascular pressures, has a limited capacity for hypertrophy. In general, PAH increases rapidly to exceed the RV-hypertrophy capacity, which leads to dilation of the right ventricle and a loss of contractility. In addition, due to the increased oxygen demand and decreased coronary perfusion, low-noise myocardial ischemia occurs, which also progressively degrades cardiac function [17,18].

**Figure 2.** Determinants of risk of pediatric idiopathic and/or familial pulmonary arterial hypertension. Adapted from [2]. RV: right ventricle; PVRI: pulmonary vascular resistance index; CI: cardiac index; PVR: pulmonary vascular resistance; SVR: systemic vascular resistance; VO$_2$: volume of oxygen uptake; 6MWD: 6-min walking test; CPET: cardiopulmonary exercise testing.

## 4. Conclusions

Idiopathic pulmonary arterial hypertension is a rare disease with a poor prognosis. The diagnosis should be made as quickly as possible, which requires good knowledge of its pathophysiology, of its clinical manifestations, and of the first-line examinations, which are mandatory. Exercise intolerance is the most frequent clinical sign, as our case illustrates. Treatment should be carried out in centers with high levels of expertise.

Although great progress has been made in recent years, notably with new studies targeting the pediatric population and that allow better care, IPAH remains a severe chronic disease that negatively affects the quality of life of affected children and carries a high burden in terms of morbidity and mortality.

**Author Contributions:** Conceptualization, T.D., M.-C.S., N.F. and M.B.; methodology, T.D., R.H. and M.-C.S.; software, T.D. and N.F.; validation, T.D., N.F., R.H., M.-C.S. and M.B.; formal analysis, T.D., M.-C.S. and M.B.; investigation, T.D., N.F. and R.H.; resources, T.D. and M.-C.S.; data curation, T.D. and R.H.; writing—original draft preparation, T.D.; writing—review and editing, M.-C.S. and M.B.; visualization, M.-C.S.; supervision, M.B.; project administration, T.D. and M.-C.S.; funding acquisition: None. All authors have read and agreed to the published version of the manuscript.

**Funding:** This research received no external funding.

**Institutional Review Board Statement:** The Ethical Review Board of The University Hospital of Liège was asked for advice and stated that a special agreement was not necessary for this kind of research.

**Informed Consent Statement:** No informed consent statement was requested in this case, in which no patient details jeopardized anonymization.

**Data Availability Statement:** No data available.

**Conflicts of Interest:** The authors declare no conflict of interest.

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
