# Peer review of "Growing up with Idiopathic Pulmonary Arterial Hypertension: An Arduous Journey"

_pediatrrep, doi:10.3390/pediatric15020026_

Round 1

Reviewer 1 Report

Many thanks to the authors for this very complete and interesting article, which reflects a topic of relevance in today's society. Although the work is considered for acceptance, it is necessary to carry out a series of previous modifications:

- The abstract must go all in a row, that is, several paragraphs must not be put inside it.

- Line 35 begins with indentation on the first line and line 42 does not, please review the entire manuscript and put it with the same structure so that it is homogeneous.

- The introduction section should be expanded to provide background information to the reader before reading it completely.

- Line 63: if this phrase refers to a subsection, it should be highlighted, for example, in italics, as it may confuse the reader.

- Line 112: the letter "t" of the word time is in red. Change.

- Lines 144-151: words written in red appear, please modify.

- All tables must be cited in the text prior to their appearance. Review the entire document.

- Line 282: words written in red appear, please modify.

- References must respect the rules of the journal, please review.

Author Response

Dear Reviewer,

thank you very much for your comments regarding our manuscript.
We took all points into consideration and have modified the manuscript accordingly.
1. Abstract: done

2. line 35 & 42: done (identations have been removed.

3. The introduction has been completed with necessary information for a better understanding of the text.

4. Line 63: it is not a sub-section, this point belongs to the same paragraph.

5. Line 112: unfortunately, the automatic underlining of text modifications is reactivated after each uploading; We hope that it is fixed now. This applies also to all words that are underlined or that appear in an other color than black. We really apologyze for that.

6. All tables and figures are cited in the text before  they appear.

7. References have been reviewed to fix the standard of the journal.

We hope that the modifications made are adequate and thank you again for your consideration and help in improving our work.

Sincerely yours

For the authors

M-C Seghaye

6.

Reviewer 2 Report

The manuscript entitled Growing up with idiopathic pulmonary arterial hypertension: An arduous trip brings an important diagnosis problem in pediatric pathology, where the early diagnosis and treatment are essential for patient's life.

The following observation have to be made

Introduction

Please provide the Ethic Committee Approval and Informed Consent signed by the child parents/tutors.

Please explain more detailed why you consider this case important to be presented and published. Why is special and different from other cases of pulmonary hypertension?

- line 65 - please explain the abbreviation of RA

Discussions

Please add the contribution of minor beta thalasemia to exercise intolerance in this case - line 136

Author Response

Dear Reviewer,

thank you very much for the comments you have made regarding our manuscript. All points were taken into consideration and the manuscript was modified accordingly.

  1. There is no need to have the approval of our Ethical Committee in our country. The statement of the president of our local EC is joined.
  2. Pointing out the importance of the case presented: This important point appears now in the introduction section, in particulmar, we would like to make the medical community aware about this pathology.
  3. Abbreviation "RA" is now explained.
  4. A comment about the possible role of b-Thalassemia is now made.

We hope that the modifications made are adequate and thank you again for your comments, your consideration and help to improve our work.

Sincerely yours

For the authors

M-C Seghaye